# Catalytic oxidation upcycling of polyethylene terephthalate to commodity carboxylic acids

Qinghai Chen[1,4], Hao Yan [1,4] ✉, Kai Zhao[1], Shuai Wang[1], Dongrui Zhang[1], Yaqian Li[1], Rong Fan[1], Jie Li[1], Xiaobo Chen[1], Xin Zhou[2], Yibin Liu [1], Xiang Feng [1] ✉, De Chen[3] ✉ & Chaohe Yang[1]

Catalytic upcycling of polyethylene terephthalate (PET) into high-value oxygenated products is a fascinating process, yet it remains challenging. Here, we present a one-step tandem strategy to realize the thermal catalytic oxidation upcycling of PET to terephthalic acid (TPA) and high-value glycolic acid (GA) instead of ethylene glycol (EG). By using the Au/NiO with rich oxygen vacancies as catalyst, we successfully accelerate the hydrolysis of PET, accompanied by obtaining 99% TPA yield and 87.6% GA yield. The results reveal that the oxygen vacancies in NiO (NiO-$O_v$) support tend to adsorb hydrolysis product TPA, preferentially ensuring the strong adsorption of EG at the Au-NiO interface. Moreover, during the EG oxidation process, the Au-NiO interface, composed of two types of structures, quasi "AuNi alloy" and NiO-$O_v$, simultaneously promote the C-H bond activation, where Ni in "AuNi alloy" exhibits an oxytropism effect to anchor the C = O bond of the intermediate, while the residual O in NiO-$O_v$ pillages the H in the C-H bond. Such Au/NiO catalyst is further extended to promote the thermal catalytic oxidation upcycling of other polyethylene glycol esters to GA with excellent catalytic performance.

The rapid rise in plastics production for market needs has caused a global waste problem[1,2]. Taking polyethylene terephthalate (PET) for example, which is the most commonly used polyester plastic with approximately 70 million tons annually for textiles and packaging, more than 80% of them are discarded or accumulated in aquatic systems and landfills[3–7]. Under the inevitable trend of circular economy, plastic reclaim is indispensable for saving non-renewable resources, thereby reducing carbon dioxide emissions[8–11]. Traditional plastic recycling strategies, represented by mechanical methods, suffer from low recycling rate and poor quality of secondary recycled products, which is also called a downcycling strategy[4,12–14]. In this scenario, chemical recycling serves as a supplementary solution of mechanical recycling to obtain high-quality monomer subunit or other chemical products via hydrolysis, glycolysis, pyrolysis, methanolysis,

hydrogenolysis or hydrosilylation processes, etc[4,13–16]. Chemicals can be recovered through chemical recycling of waste PET, avoiding the significant consumption of fossil fuels and greenhouse gas emissions in traditional production processes. Despite these efforts, the success of above strategies relies on the efficiency and selectivity of the catalysts, as well as the profitability and sustainability of the process. Therefore, exploring more diverse and sustainable chemical upcycling strategies for transforming PET into high-value chemicals is of great importance.

Compared with methanol hydrolysis and ethylene glycol hydrolysis, alkaline hydrolysis of PET into terephthalic acid (TPA) and ethylene glycol (EG) monomers is a facile approach for PET degradation due to its relatively fast depolymerization rate and mild reaction conditions[7,17–20]. However, the depolymerization of PET via hydrolysis

[1]State Key Laboratory of Heavy Oil Processing, China University of Petroleum (East China), Qingdao, China. [2]College of Chemistry and Chemical Engineering, Ocean University of China, Qingdao, Shandong, China. [3]Department of Chemical Engineering, Norwegian University of Science and Technology, Trondheim, Norway. [4]These authors contributed equally: Qinghai Chen, Hao Yan. ✉e-mail: haoyan@upc.edu.cn; xiangfeng@upc.edu.cn; de.chen@ntnu.no

suffers from the low value-added EG product and unsatisfactory depolymerization rate, which raises doubts about the economic viability of this route. Considering the abundant forms of oxygen in PET, catalytic oxidation upgrading of hydrolyzed EG into high-value glycolic acid (GA) is a promising way, which has been already realized through a two-step process of enzyme catalysis and electro-catalysis catalyzed depolymerization followed by oxidation (Fig. 1)[21–24]. Notably, glycolic acid is an important organic synthetic raw material widely used in fields such as biology and medicine and the polymer of glycolic acid monomer—polyglycolic acid (PGA) is an important green and environmentally friendly biodegradable plastic product. Unfortunately, traditional high pollution and toxic chloroacetic acid method or cyanation method limit the low-cost production of GA, which urgently needs to be supplemented by other process routes. Obviously, obtaining glycolic acid from waste PET can avoid the environmental impact of traditional production of glycolic acid, which consumes fossil resources and generates a large amount of greenhouse gas emissions. However, the low production efficiency of the two-step process may emerge as a hidden danger during the industrial scale in face of such a large scale of PET consumption. Based on this point, thermal catalytic oxidation upcycling of PET to TPA and GA via one-step strategy, named one-step process of thermal catalysis, seems to be a feasible solution. To date, noble metals (Pt, Pd and Au) have demonstrated to exhibit superior catalytic activity for EG oxidation[25–27]. Nevertheless, the glycolic acid selectivity is highly sensitive to reaction temperature and time in the cascade catalytic process, leading to the difficulty in balancing activity and selectivity[28,29]. In particular, it is challenging to ensure the occurrence of both PET depolymerization and EG oxidation, containing the multistep nature of the transformation processes, under the same reaction condition. In other words, designing efficient catalysts for EG oxidation under PET hydrolysis reaction conditions is of great importance for achieving this in situ oxidation upgrade strategy. To the best of our knowledge, thermal catalytic oxidation upcycling of PET into TPA and GA over supported catalysts is largely unexplored.

In this study, we demonstrate a simultaneous transformation of the hydrolysis of PET to TPA and EG, and the oxidation of EG to GA, over the Au/NiO catalyst with rich oxygen vacancies at 130 °C and 1 MPa $O_2$. In addition to the accelerated PET hydrolysis rate, value-added chemicals including TPA (Yield > 99%) and GA (Yield > 87%) are produced in high yields. Considering that the hydrolysis product TPA can easily cover the active sites with oxidation ability and thereby inhibit the generation of GA product during the catalytic oxidation process of PET, we leverage oxygen vacancies in NiO (NiO-$O_v$) support tend to adsorb TPA, preferentially ensuring the strong adsorption of EG at the Au-NiO interface. Meanwhile, the oxygen vacancies at interface induce the formation of quasi "AuNi alloy" and NiO-$O_v$ to promote the C-H bond activation. Finally, through continuous hydrolysis and

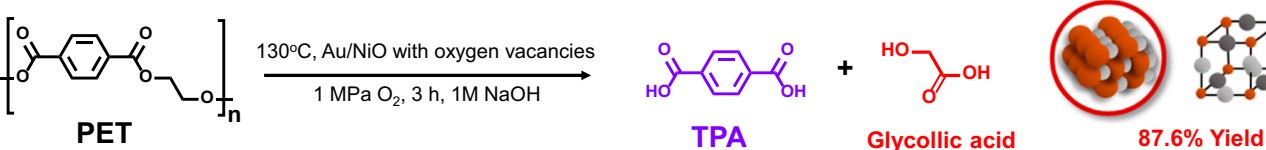

**Fig. 1 | Reported PET recycling methods and glycolic acid synthesis routes. a** Traditional process for hydrolysis and oxidation. **b** Catalytic oxidation upcycling of two-step process of enzyme catalysis and electro-catalysis catalyzed depolymerization followed by oxidation and one-step of thermal catalysis.

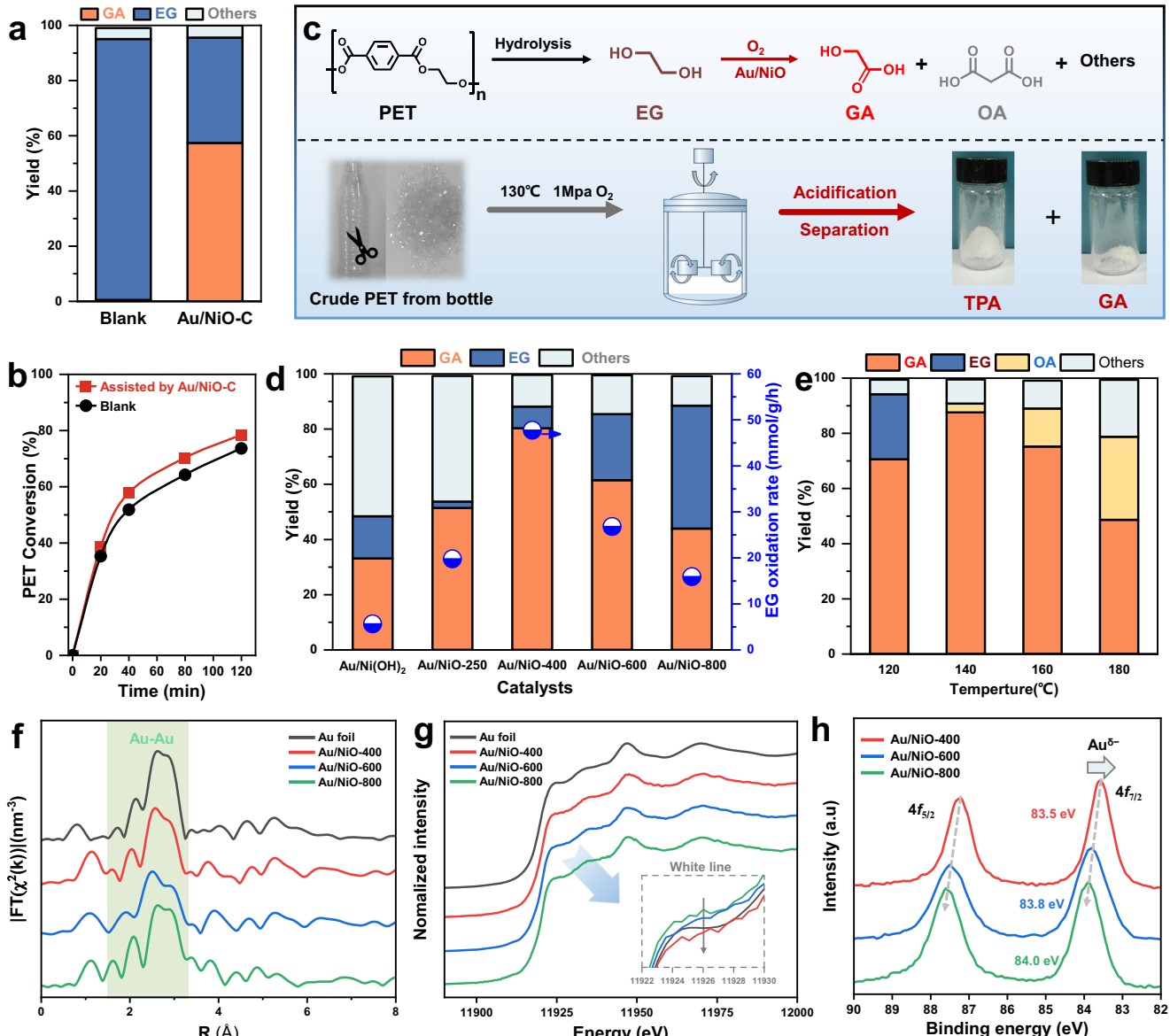

**Fig. 2 | Thermal catalytic oxidation upcycling of PET into TPA and glycolic acid (notably, all the experiment data in the figures are calculated based on the theoretical amount of EG monomer in PET). a** Traditional hydrolysis and catalytic oxidation upcycling of commercial PET. **b** Hydrolysis rate profile with/without the assistance of Au/NiO-C catalyst. **c** Catalytic oxidation upcycling of PET bottles and the subsequent separation and purification process of TPA and GA products. **d** Catalytic performance over Au/Ni(OH)$_2$ and Au/NiO-x catalysts at 130 °C. **e** The influences of reaction temperature on EG conversion and product selectivity over Au/NiO-400 catalyst. **f** Fourier transform of Au K-edge extended EXAFS oscillations. **g** First-order derivatives of Au K-edge XANES. **h** Au 4$f$ spectra of Au/NiO-400, Au/NiO-600 and Au/NiO-800 catalysts.

oxidation processes, stoichiometric productions of TPA and GA are achieved over the Au/NiO catalyst. The demonstrated strategy offers an innovative and economically viable solution for upcycling of polyethylene glycol esters to produce high-value oxygenated products with optimal atom utilization.

## Results

### Thermal catalytic oxidation upcycling of PET into TPA and glycolic acid over Au/NiO

The alkaline hydrolysis and catalytic oxidation upcycling experiments were both carried out in the batch reactor using 1 g PET as feedstocks in the 20 ml NaOH aqueous solution (Supplementary Fig. 1, 2). Figure 2a shows that the traditional alkaline hydrolysis mainly obtains 99.1% yield terephthalic acid (TPA) and 94.6% yield ethylene glycol (EG) as the main products, accompanied by a small amount of CO$_2$ as the

by-product, at 130 °C and 3 h of reaction time and 1 MPa of O$_2$ pressure (Supplementary Table 1). In contrast, thermal catalytic oxidation upcycling process, employing commercial NiO supported Au particle (Au/NiO-C) as catalyst prepared by sol immobilization method[30], can not only obtain 99.3% yield TPA, but also oxidize the depolymerized EG, resulting in an 57.4% yield of glycolic acid (measured by the molar amount of PET consumed). Moreover, the depolymerization rate of PET in this process is slightly improved compared to the traditional alkaline hydrolysis process (Fig. 2b), due to the rapid consumption of EG obtained from depolymerization, which in turn promotes the occurrence of the previous PET depolymerization process. In other words, we successfully achieved in-situ rapid upcycling of PET to TPA and high-value GA via thermal catalytic oxidation with the assistance of Au/NiO-C catalyst. Even if the commercial PET feedstock is replaced by bottle PET, Au/NiO-C also maintained excellent catalytic performance

(99.3% yield TPA and 80.3% yield GA at most). On this basis, these two products were further subjected to subsequent acidification and crystallization separation, ultimately obtaining pure TPA and GA crystals (Fig. 2c) which were confirmed by FT-IR (Supplementary Fig. 3). In addition, we conducted further amplification experiments to verify the feasibility of large-scale development. A 500 ml high-pressure reactor was employed to process 30 g PET particles in a single run, and PET was completely reacted. After separation and purification, 8.95 g of ethanol acid (yield: 75.71%) and 25.78 g of terephthalic acid (yield: 99.31%) were finally obtained (Supplementary Fig. 4).

To further optimize reaction performance, a series of NiO-x supports were synthesized by calcining Ni(OH)$_2$ obtained from deposition precipitation method at different temperatures to support Au nanoparticles, ultimately obtaining Au/NiO-x catalysts and applying it for this catalytic oxidation process. Figure 2d shows that the calcination temperature of support can significantly affect the catalytic oxidation performance of PET. Initially, Au/Ni(OH)$_2$ with surface hydroxyl saturation exhibits a sluggish reaction rate of both PET hydrolysis and EG oxidation. As Ni (OH)$_2$ gradually was calcined to NiO, Au/NiO-400 exhibits the fastest EG oxidation rate (47.74 mmol·g$^{-1}$·h$^{-1}$) and highest GA yield (81.2%), compared with other Au/NiO-x catalysts. Moreover, through secondary optimization of reaction temperature, the GA yield on Au/NiO-400 can be improved to 87.6% (Fig. 2e). Further increasing the calcination temperature can not affect the PET hydrolysis rate, but lead to a decrease in EG oxidation rate, which may due to the changes in catalyst structure caused by calcination in Au/NiO-600 and Au/NiO-800[31,32]. The above evaluation results indicate that Au/NiO catalysts are susceptible to the influence of Au-NiO interface structure, which requires further analysis of the underlying relationship between catalyst structure and catalytic performance.

The structure of the Au/NiO-x catalysts were characterized using various techniques. The physical properties of the sample are shown in the Supplementary Table 2. According to TEM and EDX elemental mapping (Supplementary Fig. 5, 6), all the Au/NiO-x catalysts exhibit a narrow distribution range of 4.5–5.0 nm Au nanoparticles, which are uniformly distributed on the NiO support accompanied by the generation of Au-NiO interface structure. Figure 2f shows the radial distribution calculated via Fourier transformation of the Au K-edge extended x-ray absorption fine structure spectroscopy (EXAFS) data. All the three catalysts exhibit an obvious Au-Au shell attributed to the metallic state of Au (~2.86 Å). Notably, the coordination number (CN) of the shell of Au-Au in Au/NiO-400 decreases to 9.0 compared to Au/NiO-600 (CN: 9.9) and Au/NiO-800 (CN: 10.7) (Supplementary Table 3). This suggests that the interaction between Au nanoparticles and the NiO-400 support leads to a slight difference in the coordination environment. The Au K-edge X-ray absorption near edge structure (XANES) shows that although the Au particles on the three catalysts exhibit metallic state, Au/NiO-400 exhibits an electron-rich state as observed from the white line peak (Fig. 2g). X-ray Photoelectron Spectroscopy (XPS) spectra (Fig. 2h) clearly demonstrate that the Au/NiO-400 exhibits the lowest binding energy of Au 4$f_{7/2}$, followed by Au/NiO-600 and Au/NiO-800, suggesting that suitable calcination temperature of support can induce the electron transfer from NiO to Au, resulting in the formation of Au$^{\delta-}$ on the surface[33–35]. Thermogravimetric analysis-mass spectrum (TG-MS) confirms that the Ni(OH)$_2$ can be completely consumed to form NiO above 300 °C (Supplementary Fig. 7)[36,37]. Low calcination temperature can lead to the presence of partial hydroxyl groups on the surface and bulk phase of NiO (Supplementary Fig. 8)[24], suppressing the connection between Au and NiO. Meanwhile, X-ray diffraction (XRD) and scanning electron microscope (SEM) in Supplementary Fig. 9 show that with the increase of the calcination temperature, the crystallinity of hexagonal phase NiO is enhanced, accompanied by the aggregation of NiO particles from 10 nm to 49.5 nm. Based on these foundations, the physical structure

changes of the NiO caused by calcination temperature need further investigation since it may exhibit a significant impact on the properties of the Au-NiO interface.

## The role of oxygen vacancies in Au/NiO for the selective oxidation process

Firstly, the electronic properties of Ni and O in NiO were investigated by XPS and DFT calculation. The XPS of Ni 2$p$ in Supplementary Fig. 10 shows that all the Au/NiO-x catalysts display typical electron absorption peaks of NiO with Ni$^{2+}$ as the main component[38–42], while the Au/NiO-400 exhibits higher binding energies of Ni$^{2+}$ (853.6 eV) than Au/NiO-600 and Au/NiO-800, confirming the electron transfer from NiO to Au. The essential reason for such surface electron reconstruction can be explained by the formation of rich oxygen vacancies in Au/NiO-400. The O 1$s$ region can be deconvoluted into three peaks: (1) adsorbed hydroxyl groups or water at ~532.0 eV (O$_I$), (2) surface oxygen vacancy at ~531.0 eV (O$_{II}$) and (3) lattice oxygen of NiO at ~529.5 eV (O$_{III}$)[31,32,41]. Figure 3a shows that Au/NiO-400 has a high oxygen vacancy content (34.38% O$_{II}$) and a minimum lattice oxygen content (56.44% O$_{III}$). In sharp contrast, the oxygen vacancy content of Au/NiO-600 and Au/NiO-800 with high calcination temperature of support gradually decreases. Low-temperature electron paramagnetic resonance (EPR) also reveals that Au/NiO-400 exhibits the strongest oxygen vacancy signal at a g-factor of 2.003, followed by Au/NiO-600 and Au/NiO-800 (Fig. 3b)[31,32]. Consistent with O1$s$ XPS and EPR analysis, the temperature programmed desorption (TPD) of O$_2$ further confirms the differences in surface oxygen vacancies among the three catalysts (Supplementary Fig. 11)[43]. Based on this point, two calculation models of Au/NiO and Au/NiO-O$_v$ were constructed to investigate the effect of oxygen vacancies on the electronic structure of catalysts (Supplementary Fig. 12). We observe that the formation of oxygen vacancy (O$_v$) in Au/NiO-O$_v$ leads to a decrease in the positive charge of adjacent Ni from 0.35 and 0.40 |e| to 0.47 and 0.73 |e| respectively (Supplementary Fig. 13). Moreover, the Au and nearby Ni at the O$_v$ of interface are in closer contact, tending to form quasi "AuNi alloy" structure. Such structure induces the changes in charge of bottom Au atoms from -0.05 |e| to -0.07 |e|, which is consistent with previous XPS results. Obviously, the enrichment of oxygen vacancies in Au/NiO-400 results in a rearrangement of the electronic structure at the Au-NiO interface, ultimately promoting the formation of Au$^{\delta-}$.

In situ EPR was further employed to probe the evolution of oxygen vacancy of Au/NiO-400 during catalytic oxidation of PET. Figure 3c shows that the symmetrical EPR peaks at $g = 2.003$ is attributed to unpaired electrons of O$_v$ in Au/NiO-400, and the intensity of this peak exhibits a progressive increase with rising reaction temperature, suggesting that oxygen vacancies become more active at high temperature. We further tested the production rate of GA at different temperatures (100–200 °C) and correlated it with the EPR signal strength of O$_v$. Interestingly, the increase in the EPR signal of O$_v$ corresponds to a proportional increase in GA production rate (Fig. 3d). Notably, TPA, as the carboxylic caid products, is generated along with the catalytic oxidation upcycling of PET. We also observe the decrease of conversion in individual EG oxidation by introducing different concentrations of TPA (Supplementary Fig. 14). We speculate that O$_v$ may play a crucial role in promoting EG oxidation to counteract the inhibitory effect of TPA during the catalytic oxidation cycle of PET. Based on this point, we conducted reaction kinetics experiments on individual EG oxidation and EG with TPA oxidation. Figure 3e shows that the reaction orders with respect to EG (n$_{EG1}$) and EG with TPA (n$_{EG2}$) rapidly decrease as the transitions of support from Ni(OH)$_2$ to NiO, while high calcination temperature of NiO can induce the increase of both n$_{EG1}$ and n$_{EG2}$ due to the reduction of oxygen vacancies. Meanwhile, n$_{EG2}$ on all catalysts is higher than n$_{EG1}$, indicating that the introduction of TPA can weaken the adsorption ability of EG over the Au-NiO interface. However, the difference in values between n$_{EG2}$ (1.12)

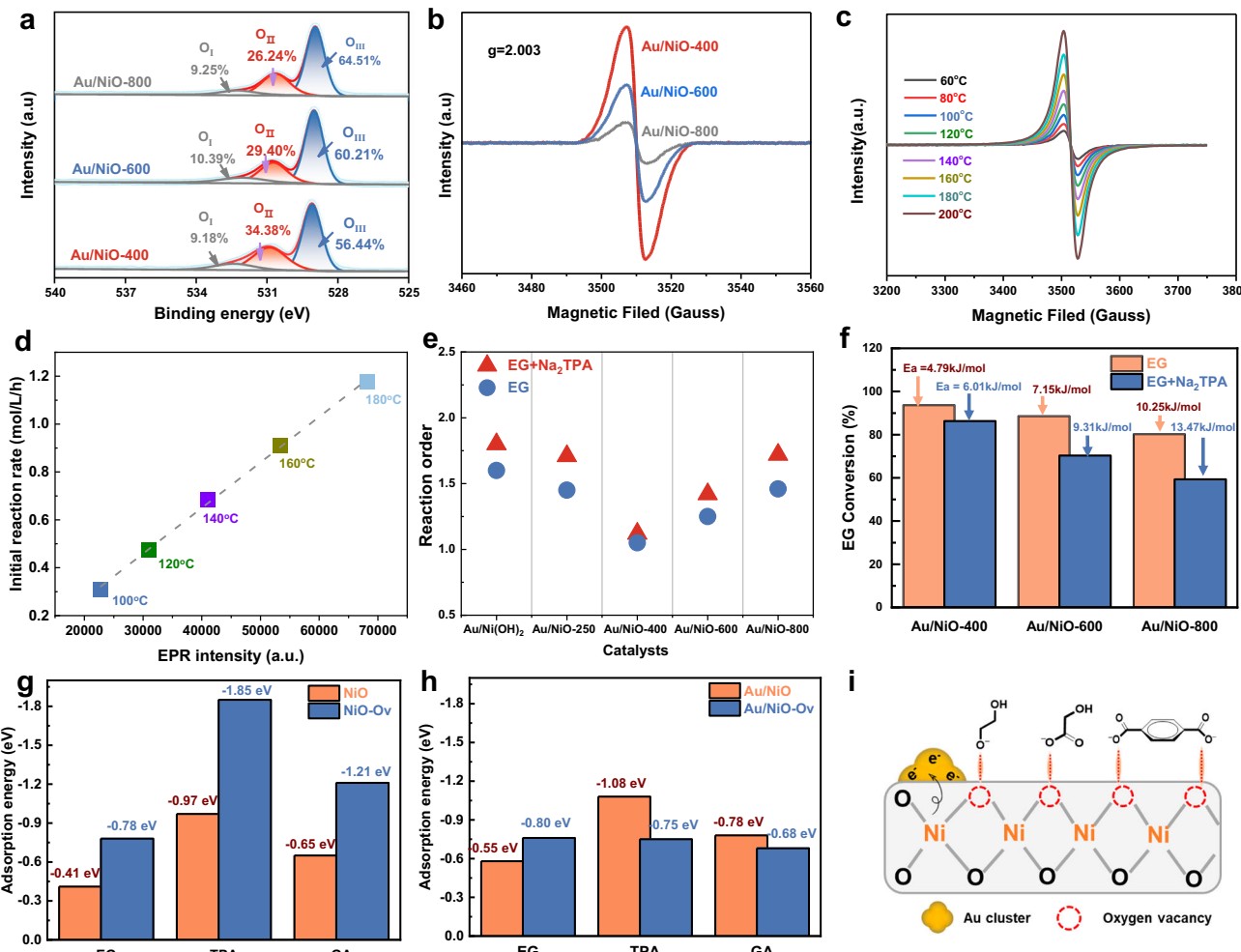

**Fig. 3 | The role of oxygen vacancies in the Au/NiO-x catalysts for the thermal catalytic oxidation upcycling of PET into TPA and glycolic acid. a** O 1s XPS spectra of the Au/NiO-x catalysts. **b** EPR spectra of the Au/NiO-x catalysts. **c** In situ EPR spectra for the detection of the evolution of oxygen vacancies on the Au/NiO-400 catalyst. **d** The linear relationship between GA initial reaction rate (mol·L$^{-1}$·h$^{-1}$) and EPR intensity. **e**, **f** The kinetic parameters (including the reaction order of EG (**e**) and activation energy (**f**)) of EG oxidation with or not Na$_2$TPA. **g**, **h** Adsorption energy of EG, GA, and TPA obtained by DFT calculation over the Au/NiO and Au/NiO-O$_v$ models. **i** Adsorption diagram of EG, GA and TPA on Au/NiO-Ov.

and n$_{EG1}$ (1.05) on Au/NiO-400 is the smallest compared with other catalysts, suggesting that sufficient oxygen vacancies may ensure the adsorption of EG under the influence of TPA. In addition, the activation energy of EG oxidation on Au/NiO-400 is only 4.79 kJ/mol, and even with the introduction of a certain concentration of TPA, the activation energy can not increase significantly (6.01 kJ/mol) (Fig. 3f). On the contrary, Au/NiO-600 and Au/NiO-800 with few oxygen vacancies not only display high activation energies for EG oxidation (7.15 and 10.25 kJ/mol, respectively), but the introduction of TPA significantly further increases this value. Moreover, the catalytic activity of Au/NiO-400 gradually decreased as the number of catalytic cycles increased (Supplementary Fig. 14). The yield of glycolic acid decreased by 37.2% compared to that of the fresh catalyst after four cycles. After calcining the spent catalyst in a nitrogen atmosphere, the catalytic activity of Au/NiO-400 was recovered due to an increase in surface oxygen vacancies. However, due to aggregation of gold particles, the glycolic acid yield was lower than that reacted with the fresher catalyst. It is worth noting that throughout all experiments, a high selectivity (>85%) for glycolic acid was obtained, indicating that the Au/NiO interface plays a crucial role as the active site for ethylene glycol conversion and further highlighting the significance of vacancies in facilitating this reaction.

The adsorption behaviour calculated by DFT calculation in Supplementary Fig. 15 and Fig. 3i show that oxygen containing species, including EG, GA, and TPA, tend to adsorb on the oxygen vacancies of NiO due to the lack of coordination O around Ni, resulting in higher adsorption energy on NiO-O$_v$ than on NiO. In particular, TPA adsorbs onto NiO and NiO-O$_v$ in a flat structure, resulting in adsorption energies as high as −0.97 and −1.85 eV respectively (Fig. 3g). Meanwhile, the formation of Au-NiO interfacial structure in the Au/NiO promotes the adsorption of EG, TPA and GA due to the synergistic adsorption effect of Au site, increasing from the adsorption energy of −0.41, −0.67 and −0.55 eV to −0.55, −0.88 and −0.78 eV respectively. However, the Au-NiO-O$_v$ interfacial structure in Au/NiO-O$_v$ only enhances the adsorption of EG (−0.80 eV), while significantly weakens the adsorption of TPA (−0.75 eV) and GA (−0.68 eV) (Fig. 3h). The quasi "AuNi alloy" structure in the Au-NiO-O$_v$ interface exhibits a lower 5 d band center (−3.07 eV) compared with that in Au-NiO interface (−1.74 eV) (Supplementary Fig. 12). On the one hand, this enhances the adsorption of H in the hydroxyl group, resulting in a slight increase in the adsorption energy of EG. On the other hand, it weakens the adsorption of similar oxygen-containing substances such as TPA by changing their adsorption configuration from planar to vertical. Inferentially, Au/NiO-400 catalysts with more oxygen vacancies tend to adsorb TPA and generated GA products onto the rich oxygen vacancies, preferentially ensuring strong adsorption of EG over the Au-NiO-O$_v$ interface. To further corroborate computational findings, a series of infrared (IR)

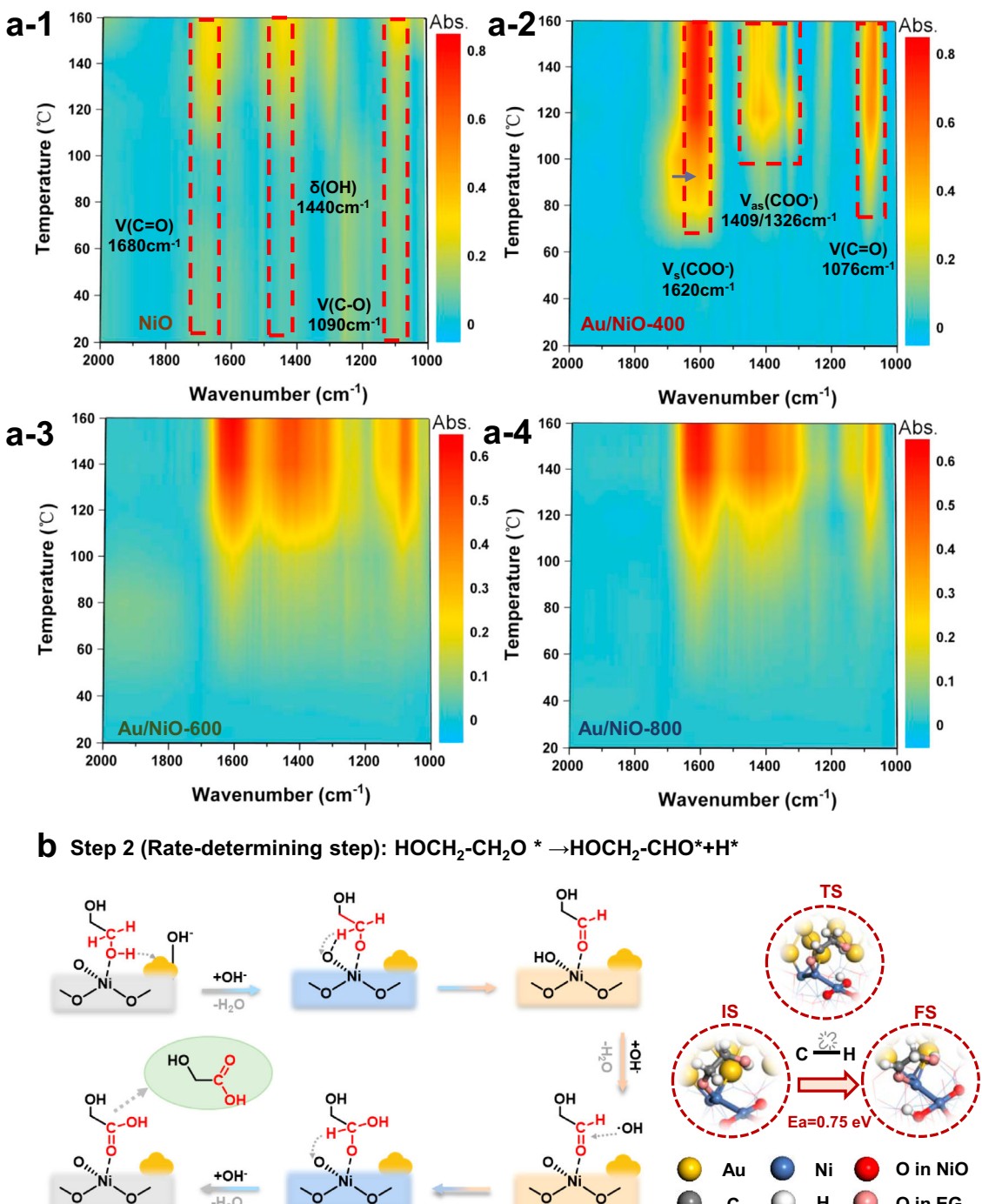

**Fig. 4 | Mechanistic investigations for EG oxidation to GA. a** In situ FT-IR spectra of EG oxidation to GA over the NiO support, Au/NiO-400, Au/NiO-600, and Au/NiO-800 catalysts. **b** Schematic diagram of the reaction mechanism for the EG oxidation to GA (the gray box represents O-H bond activation, the blue box represents C-H bond activation, and the orange box represents hydroxyl radical reaction).

tests were conducted to confirm that the introduction of $O_v$ improves the adsorption of EG at the Au/NiO interface. Supplementary Fig. 16 clearly shows that EG and TPA desorb at higher temperatures on the Au/NiO-$O_v$ surface compared to the Au/NiO surface with vacancies, confirming that the presence of vacancies enhances substrate adsorption on the catalyst surface. By comparing the adsorption behaviors of the two catalysts in actual reaction solutions (Supplementary Fig. 17), it is apparent that Au/NiO-$O_v$ adsorbs more ethylene glycol, which results in a faster conversion of EG to glycolic acid at the reaction active sites. Consequently, we have substantiated our prior hypothesis through kinetic experiments, DFT calculations, and

infrared analyses, that $O_v$ diminishes the inhibitory impact of TPA on EG oxidation, mediated by the differential adsorption strength of the substrate on $O_v$.

## Mechanistic investigations for the thermal catalytic oxidation upcycling

The reaction mechanism of EG oxidation to GA over the Au-NiO and Au-NiO-$O_v$ interface was further explored in situ fourier transform infrared reflection (in situ FTIR) and DFT calculation. The role of NiO support with rich oxygen vacancies in ethylene glycol oxidation was first investigated. Figures 4a–1 show that the intensity of the peaks at

1090 cm$^{-1}$ and 1440 cm$^{-1}$[44,45], attributed to the stretching vibration of C-O ($v$(C-O)) and the deformation vibration of −OH ($\delta$(−OH)), respectively, slightly increases with the increase of temperature. Meanwhile, the peak at 1680 cm$^{-1}$, attributed to the stretching vibration of C = O bond ($v$(C = O)) appears, indicating the formation of the intermediate of ethanol aldehyde (CH$_2$OHCHO*). Notably, no vibration peak corresponding to carboxyl group (COO$^-$) was observed. Obviously, the NiO support with oxygen vacancies exhibits weak activation ability of EG, while cannot selectively activate it into GA. Figures 4a−2 show the reaction process of EG oxidation to GA on Au/NiO-400 catalyst. Compared with NiO support, Au/NiO-400 exhibits strong symmetric stretching vibration ($v_s$(COO$^-$)) and asymmetric stretching vibration peaks of COO$^-$ functional group ($v_{as}$(COO$^-$)) in GA, located at 1620 cm$^{-1}$ and 1409/1326 cm$^{-1}$. In addition, a stretching vibration peak of the C = O bond of CH$_2$OHCHO* ($v$(C = O)) in aldehyde is observed at 1076 cm$^{-1}$. Based on the above analysis, we speculate that the oxidation pathway from EG to GA on Au/NiO-x is CH$_2$OHCH$_2$OH* (EG) → CH$_2$OHCHO* (ethanol aldehyde) → CH$_2$OHCOOH* (GA). Moreover, by comparing Au/NiO-600 and Au/NiO-800 with low oxygen vacancy content (Fig. 4a-3, a-4), we found that Au/NiO-400 exhibits highest EG oxidation activity since it displays the characteristic absorption peaks of the above oxidation process at only 60 °C.

Figure 4b shows a schematic diagram of Au/NiO-O$_v$ oxidation of ethylene glycol to produce glycolic acid obtained from DFT calculation (Supplementary Figs. 18, 19). Initially, the hydroxyl radicals generated by the interaction of O$_2$ and H$_2$O on the catalyst participate in the subsequent activation of ethylene glycol, which is consistent with previous studies. Then, EG is adsorbed on the exposed Ni over the Au-NiO-O$_v$ interface, and nearby Au$^{\delta-}$ attracts H from the O-H bond of EG with the assistance of OH$^-$. Next, the C-H bond activation of the oxygen-containing intermediate is further promoted by the interfacial structure. Notably, this step is the rate-determining step (RDS) which was proven by the kinetic isotope effect (KIE) experiment for EG oxidation (Supplementary Fig. 20), and Au/NiO-O$_v$ exhibits a lower activation energy of RDS than Au/NiO. From the configuration of transition state (TS), the Au-NiO-O$_v$ interface can be divided into two parts: quasi "AuNi alloy" and NiO-O$_v$, where Ni in "AuNi alloy" exhibits an oxytropism effect to anchor the C = O bond of the intermediate, while the residual O in NiO-O$_v$ pillages the H in the C-H bond. Such synergistic effect is the key to improving the performance of EG oxidation to GA on the Au/NiO-400 catalyst. Subsequently, the as-formed ethanol aldehyde is attacked by hydroxyl radical to generate CH$_2$OHCHOOH, which further undergoes dehydrogenation reaction to obtain GA. This process was demonstrated through free radical quenching experiments and ex-situ EPR characterization (Supplementary Table 4 and Supplementary Figs. 21, 22). Consequently, the reaction mechanism of thermal catalytic oxidation upcycling of PET into TPA and glycolic acid over Au/NiO can be inferred as follows: the depolymerization process of PET is spontaneous, while the continuous consumption of EG drives this process to occur. Meanwhile, TPA generated together with EG can preferentially adsorb on the oxygen vacancies of NiO support, ensuring the Au-NiO-O$_v$ interface to activate EG. During the EG oxidation process, the Au-NiO-O$_v$ interface is composed of two types of structures, quasi "AuNi alloy" and NiO-O$_v$, to accelerate the occurrence of RDS. Ultimately, thermal catalytic oxidation upcycling of PET into TPA and glycolic acid is achieved and accelerated by the Au/NiO catalysts with rich oxygen vacancies.

## Economics and environment impact of the thermal catalytic oxidation upcycling strategy

To further demonstrate the advantages of the one-step oxidation strategy in PET treatment, we conducted a costs and product yields analysis for the four recycling routes using the established techno-economic assessment model (Supplementary Fig. 23)[46], based on an annual treatment capacity of 100,000 tonnes of PET. Figure 5a shows

that conventional alkaline hydrolysis of PET for EG production can obtain 450.08$/ton of gross profit, where the recycled TPA constitutes over 80% of the revenue. Compared with alkaline hydrolysis, two-step processes of emzyme catalysis and electro-catalysis for the upcycling of PET display higher revenue due to the production of high-value GA (4879$/t) instead of EG (1260$/t). Additionally, the generation of hydrogen during the electrocatalytic oxidation of ethylene glycol enhances the process's economic viability. However, the electro-catalytic and PET pretreatment equipment necessitate greater capital expenditure. Notably, thermal catalytic oxidation upcycling of PET into TPA and GA over Au/NiO catalyst exhibits both low cost and high revenue, resulting in the large gross profit (1508.85$/ton PET) (Supplementary Table 5–8). Figure 5b shows that in comparison with electrocatalytic and enzyme-catalyzed processes (treating 1 g of PET)[21,22], the one-step oxidation strategy exhibits significant advantages, including higher yields of ethanoic acid, reduced formation of by-products, shorter reaction times, and simplified operating procedures (Supplementary Table 9). On the one hand, the time-consuming enzymatic treatment is inefficient for addressing the escalating contamination from discarded PET waste. On the other hand, although the electrocatalytic process enables efficient catalysis from EG to GA, it depends on the alkaline hydrolysis pre-treatment of PET, which compromises treatment efficiency originated from its two-step operation process. Even though simultaneous implementation of PET pre-treatment and electrocatalytic processes in industrial production can enhance efficiency, it incurs higher input costs and energy consumption compared to one-step conversion. These findings highlight that our one-step oxidation strategy holds substantial economic potential for waste PET upgrading towards GA.

Life cycle assessment was also applied to analyze the greenhouse gas emissions generated by different chemical recycling pathways of waste PET. Due to the widespread use of sorted PET flakes as raw materials in current chemical recycling, the system boundary of this LCA is set from gate to gate (Supplementary Fig. 24). The cut-off method was introduced into PET recycling to simplify the analysis. The functional unit is 1 kg PET flake after sorting and shredding. The production process data of tap water, electricity, steam, and basic chemicals were obtained from ecoinvent v3.10 database. Figure 5c shows that compared to direct incineration or landfill treatment of PET (10.51 kg CO$_2$-eq/kg PET)[3,47,48], converting waste PET into chemicals can greatly reduce greenhouse gas emissions and non-renewable energy use. Moreover, TPA and GA recovered from waste PET can avoid the significant consumption of fossil resources and a large amount greenhouse gas emissions caused by conventional production. As a result, the oxidation strategy for treating 1 kg of waste PET can avoid the use of 37.09 MJ of fossil sources and reduce the emission of 2.25 kg CO$_2$-eq. In addition, the NREU and GWP recovered by one-step oxidation route are reduced by 15.09 MJ/kg PET and 0.91 kg CO$_2$-eq/kg PET compared to the alkaline hydrolysis process. Moreover, in comparison to enzymatic catalysis, the one-step method offers a reduction in emissions, attributed to its higher yield of glycolic acid and simpler separation process. Meanwhile, we also compared the depolymerization oxidation process with other PET recycling processes, including glycolysis pyrolysis and methanolysis (Supplementary Table 10), and the results are gratifying. It is found that the depolymerization oxidation process for preparing glycolic acid is superior to the existing recycling processes in terms of energy saving and emission reduction. Obviously, this method provides a feasible environmental approach for recycling waste PET. Furthermore, we successfully applied the one-step strategy for treating other polyesters with EG monomer, such as PFE (Polyethylene 2,5-furandicarboxylate), PEN (Polyethylene naphthalate two formic acid glycol ester) and PEA (Poly (ethylene adipate)), obtaining efficient catalytic oxidation with high GA yields (>80%) (Fig. 5d). This demonstrates that our oxidation strategy holds universal applicability for treating polyester plastics.

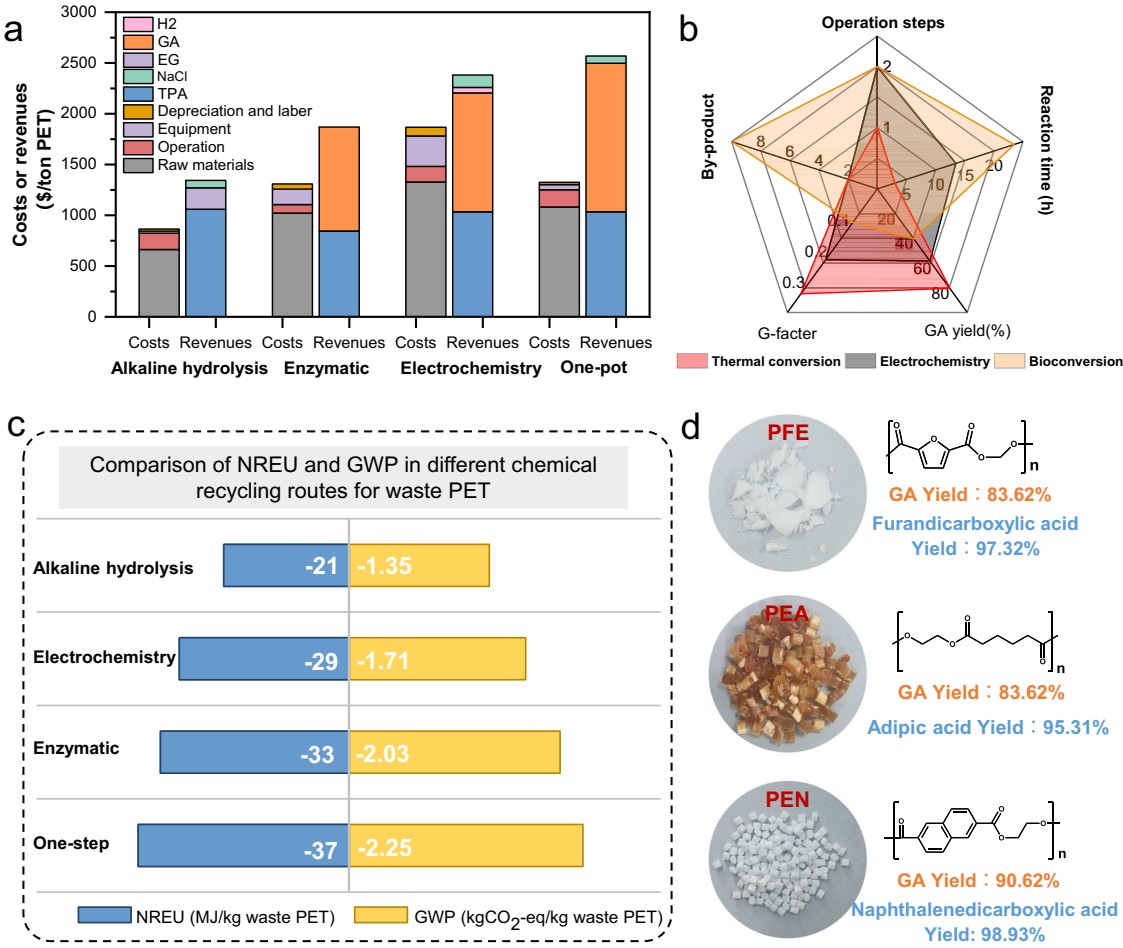

**Fig. 5 | Life cycle assessment for the thermal catalytic oxidation upcycling of PET. a** Comparison of the revenues and costs of four routes toward GA production (100,000 tons PET wastes consumption per year). **b** Comparison of the viabilities of thermal conversion, electrochemistry, and bioconversion. **c** Comparison of non-renewable energy use (NREU) and global warming potential (GWP) in different chemical recycling routes for waste PET. **d** Catalytic performance for thermal catalytic oxidation upcycling of PFE (Polyethylene 2,5-furandicarboxylate), PEN (Polyethylene naphthalate two formic acid glycol ester) and PEA (Poly (ethylene adipate)) into GA and another monomer acid over Au/NiO-400.

## Discussion

In summary, we reported a one-step strategy for the thermal catalytic oxidation upcycling of PET to TPA and high-value GA, leveraging the catalytic oxidation performance of Au-NiO interface from the Au/NiO catalyst with rich oxygen vacancies. The advantages of this process are that it not only accelerates the hydrolysis of PET, but also promotes the oxidation of EG to GA, obtaining value-added chemicals including TPA (Yield > 99%) and GA (Yield > 87%) in high yield. Notably, the oxygen vacancies of the Au/NiO catalyst play an important role in the EG oxidation process. On the one hand, the oxygen vacancies in NiO support tend to adsorb TPA, preferentially ensuring the strong adsorption of EG at the Au-NiO interface. On the other hand, the oxygen vacancies at interface induce the formation of quasi "AuNi alloy" and NiO-O$_v$, where Ni in "AuNi alloy" exhibits an oxytropism effect to anchor the C = O bond of the intermediate, while the residual O in NiO-Ov pillages the H in the C-H bond. The catalytic system is also effective in converting polyethylene naphthalate two formic acid glycol ester, poly(ethylene adipate), and polyethylene 2,5-furandicarboxylate into carboxylic acid monomers and GA in high yields. In addition, from an economic perspective, the strategy reported in this work suppresses other upcycling method of PET reported in the current literature. This approach also opens up new opportunities for the design of catalytic oxidation systems that enable the highly efficient utilization of waste materials, thereby paving to a more sustainable and environmentally friendly future.

## Methods

### Catalyst preparation

In this paper, the typical preparation process of nickel (NiO) oxide support was as follows: 2.5 g Ni(NO$_3$)$_2$·6H$_2$O was weighed and dissolved in 100 ml deionized water, and stirred at room temperature for 10 min. Ammonia solution (0.5 mol·L$^{-1}$) was added slowly under stirring to adjust pH = 9 to get Ni(OH)$_2$. After filtration, washing several times and dried at 80 °C for 6 h, a series of NiO-x samples (x = 400, 600, 800) were finally prepared by controlling the calcined temperature.

The Au/NiO-x catalysts are prepared by a sol immobilization method. In a typical process, 0.5 ml HAuCl$_4$ (20 mg·ml$^{-1}$) and PVA (mass of PVA: mass of Au = 1:1) were added into 100 ml deionized water. A freshly prepared NaBH$_4$ solution (0.1 M, NaBH$_4$: Au = 4:1, mole/mole) was added to the solution immediately, forming an orange-yellow gold sol. Then, 1 g support material (NiO-x) was added to the colloidal solution, stirred for 2 h, filtered and washed with 500 ml of deionised water to remove Na$^+$, BH$_4^-$ and Cl$^-$. After drying overnight (80 °C), the sample was calcined to remove the capping surfactant under air atmosphere at 250 °C for 2 h.

### Catalytic test

PET depolymerisation and oxidation processes were investigated in a 50 mL autoclave. In a typical process, 1 g commercial PET granule, 20 ml 1 M NaOH aqueous solution and 0.1 g catalyst were added in the

autoclave, followed by 1 Mpa $O_2$ before the reaction. At the end of the reaction, the filtrate which had been filtered off the catalyst and unreacted PET was acidified with hydrochloric acid. The liquid product filtered off terephthalic acid was analyzed by high performance liquid chromatography (HPLC) with refractive index (RID-10A) and UV detectors (Shimadzu LC-20AT). The Rezex ROAO rganic Acid H$^+$ (8%) was used as the chromatographic column in the mobile phase (0.005 M $H_2SO_4$). A schematic of this process is also shown in Supplementary Figs. 1, 2. The PET conversion, selectivity, yield to glycolate, and carbon balance were calculated with the following equations:

$$PET\ conversion(\%) = \frac{m_{PET.converted}}{m_{PET.addition}} \times 100\% \tag{1}$$

$$TPA\ yield(\%) = \frac{m_{TPA.yield}}{\frac{m_{PET.addition}}{192} \times 166} \times 100\% \tag{2}$$

$$GA\ yield(\%) = \frac{N_{GA.yield}}{\frac{m_{PET.addition}}{192}} \times 100\% \tag{3}$$

$$EG\ yield(\%) = \frac{N_{EG.yield}}{\frac{m_{PET.addition}}{192}} \times 100\% \tag{4}$$

$$M_{(TPA)} = 166 g mol^{-1}, M_{(repeated\ links\ in\ PET)} = 192 g mol^{-1}$$

## Theory calculations

All DFT calculations presented in this work were conducted utilizing the DMol$^3$ module of Materials Studio 2019 software. The generalized gradient approximation (GGA) and the Perdew-Burke-Ernzerhof (PBE) functional were selected to describe exchange-correlation energy. The double-numeric polarized basis set (DNP) with the effective core potentials (ECP) method was employed for electron treatment. The $3 \times 3 \times 1$ Monkhorst-Pack grid was used for sampling the Brillouin zone. To guarantee the high accuracy of the calculation results, allowable convergence value for total energy, displacement and gradient are $1.0 \times 10^{-5}$ Ha, 0.005 Å and 0.002 Ha/Å, respectively. The LST/QST method was utilized appropriately to search for transition states, and the rationality of these states was subsequently confirmed through frequency calculations and transition state optimization. For the computational model, The NiO(100) surface was constructed to four layers ($3 \times 2 \times 4$) with 48 atoms, of which the two top layers were relaxed, while the two bottom layers were fixed during the calculation. And a supported $Au_8$ clusters are loaded on the surface to form Au/NiO(100) structure. One O atom in the surface Ni-O-Ni structure was removed to produce oxygen vacancy($O_v$).

The adsorption energy is defined as $E_{ads} = E_{Au/NiO+Mol} - E_{Au/NiO} - E_{Mol}$, where $E_{Au/NiO+Mol}$, $E_{Au/NiO}$ and $E_{Mol}$ are the total energy of the calculation system, bare catalyst, and free molecule respectively. The free energy at specific temperature were calculated by the formula $G = E_{stru} + G_{corr}$, where $E_{stru}$ is the total energy of the structure itself and $G_{corr}$ is the free-energy correlation with the zero-point energy(ZPE) included at the specific temperature. All the energies were corrected by ZPE.

## Characterizations

X-ray diffraction patterns (X'pert PRO MPD diffractometer, CuKα) are used to analyze the crystal structure and surface properties of catalysts. Scanning electrode microscope (SEM) images were recorded by a Hitachi S-4800. Transmission electron microscopy (TEM) and high angle annular dark-field scanning transmission electron microscopy (HAADF-STEM) were performed on the JEOL JEM-2100F to determine the size of Au nanoparticles. Metal contents in catalysts were determined by ICP-OES on the VARIAN 720-ES. Raman spectra were

performed on Raman LabRAM HR Evolution 325–1000, under 325 nm laser lines. X-ray photoelectron spectra (XPS) were measured on the Thermo ESCALAB 250Xi. Temperature-programmed desorption (TPD) was tested on the Micromeritics AutoChem II 2920. The sample (0.1 g) was pretreated with 60 mL·min$^{-1}$ He at 200 °C for 1 h, and was conducted under a stream of 5 vol% $O_2/N_2$ mixture, after filled and adsorbed for 1 h, and then the TCD signal was recorded from 50 to 700 °C (10 °C·min$^{-1}$) under the flow of He (40 mL·min$^{-1}$). X-ray absorption spectroscopy (XAS) was measured on the Advanced Photon Source at shanghai (BL14B2 of SPring-8). The ATHENA module was employed to deal the data of X-ray absorption near edge structure (XANES) and extended X-ray absorption fine structure spectroscopy (EXAFS) (Supplementary Fig. 25).

Electron paramagnetic resonance (EPR) spectra were measured on a Bruker EMX-6/1 spectrometer at 100 K. For the in situ EPR experimental, 10 ml mixed solution of PET and sodium hydroxide were added to the in situ cell (1gPET + 1 M NaOH + 0.1 g Au/NiO-400). Under purged pure oxygen (100 ml·min$^{-1}$) conditions, the temperature was ramped up by a Bruker EMX plus continuous flow temperature control system and the sample EPR spectra is collected between 2400 and 3600 G after reaching the set temperature. In situ Fourier transforms infrared spectra (in situ FTIR) of ethylene glycol alkaline solution was obtained on the Thermo Scientific Nicolet iS50 with the 4 cm$^{-1}$ resolution. Specifically, the catalyst in the suit cell was first purged with $N_2$ (25 ml·min$^{-1}$) for 1 h at 100 °C, and then 20 µL of a basic solution of ethylene glycol (0.2 M EG + 1 M NaOH) was added to the catalyst surface. Nitrogen was switched to 10% $O_2$ (25 ml·min$^{-1}$) and the spectra were recorded at room temperature to be used as background. The spectra were recorded at different reaction temperatures under a set ramp-up programme.

## Life-cycle assessment

The LCA analysis followed the ISO standard series 14040 and was conducted using OpenLCA 2.1.1. The aim is to compare the environmental impact of different PET chemical recycling routes during the recycling process using two indicators: non-renewable energy use (NREU) and global warming potential (GWP). The function units is 1 kg of waste PET. The boundary of the system is from gate to gate, mainly involving PET reactions, product separation, and waste disposal. Simultaneously, the cut-off method has been applied in PET recycling treatment. The cut-off principle indicates that discarded PET is considered waste, without considering previous environmental impacts. The detailed data and assumptions can be found in Supplementary Information.

## Data availability

Source data are available from the corresponding author upon reasonable request.

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

## Acknowledgements

Our work was supported by the National Natural Science Foundation of China Outstanding Youth Foundation (22322814), the National Natural Science Foundation of China (22478429) and the Natural Science Foundation of Shandong Province (ZR2023YQ009). This work was also

supported by User Experiment Assist System of SSRF, BL14W, 113SSW and 16U1.

## Author contributions

D.C. and H.Y. conceived this work. H.Y. wrote the paper. Q.C. performed the experiments, collected the data and wrote the paper. K.Z. conducted the evaluation experiment and helped with the characterization analysis. X.Z. and D.Z. performed the economic and technical evaluation of the project. X.C., Y.B.L., and C.Y. helped with data analyses and discussions. X.F. and D.C. supervised and directed the project, assisted the XAS characterization, and revised the paper. Y.Q.L. and R.F. helped with DFT calculation. S.W. and J.L. helped with XAS characterization analysis. All authors contributed to editing the paper.

## Funding

 Olavs Hospital - Trondheim University Hospital).

## Competing interests

The authors declare no competing interests.
