## [Transparent Peer Review file · Nature Communications]

Catalytic Oxidation Upcycling of Polyethylene Terephthalate to Commodity Carboxylic Acids

Corresponding Author: Professor De Chen

Version 0:

Reviewer comments:

Reviewer #1

(Remarks to the Author)

The manuscript submitted by Chen and coworkers entitled "Catalytic Oxidation Upcycling of Polyethylene Terephthalate to Commodity Carboxylic Acids" describes a one pot depolymerization of PET to give terephthalic acid and glycolic acid using heterogeneous catalysis based on Ni and Au.

The approach to generate terephthalic acid and glycolic acid from PET is not novel, however the one-pot conversion proposed is; the yields of both products are higher than state-of-the art in chemical conversion, however not using biocatalysis. The manuscript can indeed become suitable for publication in Nat Commun. However, in the present state, the manuscript is not suitable and this reviewer suggests major revisions.

Introduction:

For PET, mechanical cascading is actually rather implemented where bottles become textiles for instance. The need for chemical recycling is driven by the fact that after cascading, the fiber eventually loses its mechanical properties; the demand of high quality fiber is high. Thus, chemical recycling is not an alternative but should be considered as an extension of the mechanical recycling. This would not change the story-line of the manuscript, but would need to be explained.

State-of-the art covers hydrolysis, however, glycolysis and alcoholysis are only covered briefly and usually are superior to hydrolysis as no neutralization step is required. Even though glycolysis would not be appropriate in generating glycolic acid, alcoholysis would. Did the authors consider this strategy?

The strategy of generating the glycolic acid is nice, and the authors are recommended to give some motivation why it would be noteworthy to generate this compound.

Results and discussion

The results are impressive, however, there is always a challenge when reactions do not give 100% yield. How would the glycolic acid be separated from the EG and others? It seems as the product mixture is only analyzed chromatographically. How was this handled in the TEA and LCA? It seems like this is not discussed at all.

The authors should demonstrate the isolation and get accurate mass balances for these events, and perform the TEA and LCA based on this.

The authors claim that the first dehydrogenation of EG to the corresponding aldehyde is the rate-limiting step. This should be verified by KIE, that can easily be performed.

Discussion

It would be worthwhile to compare the proposed method to other upcycling technologies for PET, such as:

Waste Manag. Res. 2009, 27 (8), 763–772
Resour. Conserv. Recycl. 2010, 54 (12), 1241–1249

Reviewer #2

(Remarks to the Author)

The article by Chen and collaborators presents a significant advancement in the field, describing a one-step tandem strategy to achieve the thermal catalytic oxidation upcycling of PET to terephthalic acid (TPA) and high-value glycolic acid (GA) over a well-designed Au/NiO catalyst with rich oxygen vacancies. The method, which has been successfully applied to upcycle other polyethylene glycol esters to GA, demonstrates excellent catalytic performance. The manuscript and the results reported are not only interesting but also hold great potential for publication in Nat Comm after addressing the following points.

1. On lines 188-190 and line 268 the author supposed "Inferentially, the Ov plays a crucial role in promoting EG oxidation against the inhibitory effect of TPA during the catalytic oxidation upcycling of PET (Fig. 2f)" and "TPA generated together with EG can preferentially adsorb on the oxygen vacancies of NiO support". It is recommended to conduct the in-situ IR characterization with TPA addition to verify these conclusions.
2. On lines 263-265, the author supposed "Subsequently, the as-formed ethanol aldehyde is attacked by hydroxyl radical to generate CH₂OHCHOH, which further undergoes dehydrogenation reaction to obtain GA." It is recommended to conduct an in-situ EPR test on the role of hydroxyl radicals and check whether there are other organic radical intermediates involved in the formation of GA.
3. The 1g-rated experiment may not be sufficient to demonstrate the potential of the upgrade from PET to TPA and GA. Thus, it is crucial to conduct a larger experiment or use flowing air or O₂ to illustrate the sustainability and scalability of this process.
4. Catalytic stability is very important for real applications. I am intrigued to see if the catalyst can consistently deliver its catalytic performance over 3-5 cycles.
5. The ref part contains some minor mistakes. For example, "Acs Catal" should be written as "ACS Catal".

Version 1:

Reviewer comments:

Reviewer #1

(Remarks to the Author)

The authors have taken great care in revising the manuscript and this reviewer supports publication.

Reviewer #2

(Remarks to the Author)

In this revised version, the authors have addressed most of my concerns and points related to the catalytic role and the structure-performance relationship of the Au/NiO-Ov. I only have two minor suggestions for this version, and have no doubt in recommending the publication after minor revision.

1. There is no spectra of pristine catalyst when the authors conducted FTIR spectra of adsorption of EG and TPA on catalyst surfaces. It should be added for better comparison.
2. It should not be named as "in-situ EPR" for their EPR characterization. From their description, it is better to be called "ex situ".

Responses to Reviewers' Comments

Dear Reviewers,

We greatly appreciate the opportunity to revise our manuscript. The insightful suggestions and comments from the reviewers have deepened our understanding of the one-step depolymerization and oxidation reaction mechanism of PET. We have carefully considered all comments and supplemented the manuscript with additional experimental evidence to address the queries. We wish that these revisions will be satisfactory, and that the corrections will be well received.

Reviewer #1:

The manuscript submitted by Chen and coworkers entitled “Catalytic Oxidation Upcycling of Polyethylene Terephthalate to Commodity Carboxylic Acids” describes a one pot depolymerization of PET to give terephthalic acid and glycolic acid using heterogeneous catalysis based on Ni and Au.

The approach to generate terephthalic acid and glycolic acid from PET is not novel, however the one-pot conversion proposed is; the yields of both products are higher than state-of-the art in chemical conversion, however not using biocatalysis. The manuscript can indeed become suitable for publication in Nat Communication. However, in the present state, the manuscript is not suitable and this reviewer suggests major revisions.

Major concerns:

1. Introduction:

1) Chemical recycling is not an alternative but should be considered as an extension of the mechanical recycling. This would not change the story-line of the manuscript, but would need to explained.

2) And state-of-the art covers hydrolysis, however, glycolysis and alcoholysis are only covered briefly and usually are superior to hydrolysis as no neutralization step is required. Even though glycolysis would not be appropriate in generating glycolic acid, alcoholysis would. Did the authors consider this strategy?

3)The strategy of generating the glycolic acid is nice, and the authors are recommended to give some motivation why it would be noteworthy of to generate this compound.

Response:

Thanks for the reviewer's comment. We agree with a more accurate explanation of the chemical recovery process, the selection of technical routes, and the willingness to choose glycolic acid as the target product in the introduction

1) Mechanical recycling is currently the main form of PET recycling, which is limited by the cleavage of some chemical bonds and the introduction of some impurities during the heat treatment process, resulting in a downgrade in the use of recycled PET. After several cycles, it is difficult to maintain basic performance, and ultimately landfilling or incineration treatment will undoubtedly cause pollution to the environment. The use of chemical recycling methods for this part of the material can further extend the life cycle of PET. Whether it is recycling or high-value utilization, transforming environmental pollutants into new products and turning waste into treasure is an important measure for circular economy. Thank you for your valuable suggestion. We have made some modifications to the introduction based on your suggestions to avoid any ambiguity.

2) As you suggested, alcoholysis (ethylene glycol/methanol, etc.) method can avoid the use of acid and alkali, thereby reducing the corresponding wastewater treatment and environmental pollution. However, using methanol or ethylene glycol as solvents can be difficult to avoid solvent oxidation in an oxygen atmosphere, resulting in a large amount of low value by-products such as formic acid and various esters. This method not only leads to a decrease in the conversion rate of glycolic acid, but also brings great difficulties to the separation of the target product. After comprehensive consideration, the use of alkaline depolymerization can achieve rapid depolymerization of PET under mild conditions (<180 °C), and the alkaline environment helps stabilize glycolic acid and avoid excessive oxidation.

3) Notably, glycolic acid is an important organic synthetic raw material widely used in fields such as biology and medicine. The market price of glycolic acid crystals is as high as \$1497 per ton. Moreover, the polymer of glycolic acid monomer - polyglycolic acid (PGA) is an important green and environmentally friendly biodegradable plastic product, while its high market price limits its large-scale application. Meanwhile, the traditional acid hydrolysis

production process of hydroxyacetonitrile has high toxicity and cost, which limits the large-scale production of glycolic acid. Based on these foundations, using one pot chemical recycling strategy of PET to produce glycolic acid can achieve low-cost and large-scale production of glycolic acid, thereby promoting the large-scale promotion of degradable polyglycolic acid (PGA). We have made some modifications to the introduction based on your suggestions.

“Traditional plastic recycling strategies, represented by mechanical methods, suffer from low recycling rate (<10%) and poor quality of secondary recycled products, which is also called a downcycling strategy^{4, 12-14}. In this scenario, chemical recycling serves as a supplementary solution of mechanical recycling to obtain high-quality monomer subunit or other chemical products via hydrolysis, glycolysis, pyrolysis, methanolysis, hydrogenolysis or hydrosilylation processes, etc^{4, 15-18}.” [Page 2, Line 37-39]

“Compared with methanol hydrolysis and ethylene glycol hydrolysis, alkaline hydrolysis of PET into terephthalic acid (TPA) and ethylene glycol (EG) monomers is a facile approach for PET degradation due to its relatively fast depolymerization rate and mild reaction conditions^{7, 19-22}. However, the depolymerization of PET via hydrolysis suffers from the low value-added EG product and unsatisfactory depolymerization rate, which raises doubts about the economic viability of this route. Considering the abundant forms of oxygen in PET, catalytic oxidation upgrading of hydrolyzed EG into high-value glycolic acid (GA) is a promising way, which has been already realized through a two-step process of enzyme catalysis and electrocatalysis catalyzed depolymerization followed by oxidation²³⁻²⁶. Notably, glycolic acid is an important organic synthetic raw material widely used in fields such as biology and medicine and the polymer of glycolic acid monomer - polyglycolic acid (PGA) is an important green and environmentally friendly biodegradable plastic product. Unfortunately, traditional high pollution and toxic chloroacetic acid or cyanation methods limit the low-cost production of GA, which urgently needs to be supplemented by other process routes. Obviously, obtaining glycolic acid from waste PET can avoid the environmental impact of traditional production of glycolic acid, which consumes fossil resources and generates a large amount of greenhouse gas emissions. However, the low production efficiency of the two-step process may emerge as a hidden danger during the industrial scale in face of such a large scale of PET consumption. Based on this point, thermal catalytic oxidation upcycling of PET to TPA and GA via one-step

strategy, named one-step process of thermal catalysis, seems to be a feasible solution.” [Page 2-3, Line 45-66]

2. Results and discussion: The results are impressive, however, there is always a challenge when reactions do not give 100% yield. How would the glycolic acid be separated from the EG and others? It seems as the product mixture is only analyzed chromatographically. How was this handled in the TEA and LCA? It seems like this is not discussed at all.

Response:

This is a very good question, as the residual ethylene glycol and other by-products poses a great challenge to the separation of products. Through our analysis and experiments, we have achieved the separation of complex products by using relevant operations such as concentration, filtration, and crystallization. The specific operation process is shown in the following:

Supplementary Figure 2. The process of product separation and purification.

Firstly, the reaction mixture is filtered to separate the catalyst and unreacted PET. Subsequently, hydrochloric acid (2M HCl) is added to the filtrate for acidification, adjusting the pH to 2~3 to precipitate TPA. After filtering out the formed TPA precipitate, an aqueous solution containing NaCl, ethylene glycol (EG), and glycolic acid (GA) is obtained. The filtrate is then concentrated, allowing glycolic acid and sodium chloride to gradually crystallize and precipitate as water evaporates. After it is concentrated into a certain extent, filtering the concentrated solution can separate ethylene glycol (L) from the solid product. Due to the good solubility of glycolic acid in ethanol while NaCl is almost insoluble in ethanol, glycolic acid and NaCl were separated by ethanol extraction. Finally, glycolic acid is crystallized out in ethanol, and after filtration and drying, glycolic acid crystals are obtained, while ethanol is recycled as the mother liquor.

In industry, corresponding ethanol purification processes are adopted for different synthesis methods of glycolic acid. Crystallization and recrystallization processes are used to purify the crude ethanol acid produced by chloroacetic acid hydrolysis method, and finally obtain sodium chloride and glycolic acid crystals.

Based on your suggestion, we have further revisited and retested the separation process, along with reassessing LCA and TEA. The outcomes are also promising. Utilizing an established glycolic acid separation technique, we effectively separated and purified glycolic acid from the reaction mixture (see Supplementary Fig 2). An engineering economic evaluation was conducted on the one-step depolymerization oxidation process after adding a separation process. The traditional economic evaluation model (Supplementary Fig. 23) was used to estimate the production cost and product revenue of the entire process (based on the final product yield). Thanks to the high value of glycolic acid and the low cost of the one-step method, the one-step depolymerization oxidation process can generate a revenue of \$1,508.85 per ton of PET processed, overcoming the poor economic performance of the traditional alkaline hydrolysis process and significantly higher than the enzymatic catalysis (560.99\$/ton PET) and electrocatalysis (879.82\$/ton PET) treatment processes. Enzymatic catalysis is limited by complex separation processes, resulting in lower yields of TPA and EG from PET, while the high cost of bacterial culture media leads to a lower revenue from the enzymatic catalysis process of preparing glycolic acid from PET. Although electrocatalysis has a good product yield,

the increase in preprocessing equipment and the high cost of large electrolysis equipment require higher capital investment, thereby reducing the overall economic efficiency of the process.

The evaluation of the integrated separation process confirms that the single-step depolymerization oxidation is superior in energy savings and emission reductions, primarily because existing glycolic acid production methods (such as chloroacetic acid hydrolysis and formaldehyde carbonylation) lead to substantial natural resource depletion and CO₂ emissions. Processing 1kg of PET via a single-step depolymerization oxidation can decrease resource consumption by 37.09MJ and CO₂-eq emissions by 2.25kg, outperforming enzymatic/electrocatalytic processes in energy and carbon emission reduction. Thank you once again for your thoughtful comments and invaluable suggestions, which have helped us identify our shortcomings and improve this work through necessary revisions.

Supplementary Figure 23. The cost and income for calculating techno-economic analysis of thermal catalytic oxidation upcycling of PET.

“To further demonstrate the advantages of the one-step oxidation strategy in PET treatment, we conducted a costs and product yields analysis for the four recycling routes using the established techno-economic assessment model (Supplementary Fig. 23)⁴⁸, based on an annual treatment capacity of 100,000 tonnes of PET. Fig. 4a shows that conventional alkaline hydrolysis of PET for EG production can obtain 450.08\$/ton of gross profit, where the recycled TPA constitutes over 80% of the revenue. Compared with alkaline hydrolysis, two-step processes of enzyme catalysis and electro-catalysis for the upcycling of PET display higher revenue due to the production of high-value GA (4879\$/t) instead of EG (1260\$/t). Additionally, the generation of hydrogen during the electrocatalytic oxidation of ethylene glycol enhances

the process's economic viability. However, the electrocatalytic and PET pretreatment equipment necessitate greater capital expenditure. Notably, thermal catalytic oxidation upcycling of PET into TPA and GA over Au/NiO catalyst exhibits both low cost and high revenue, resulting in the large gross profit (1508.85\$/ton PET) (Supplementary table 5-8).” [Page 16, Line 304-315]

“Life cycle assessment was also applied to analyze the greenhouse gas emissions generated by different chemical recycling pathways of waste PET. Due to the widespread use of sorted PET flakes as raw materials in current chemical recycling, the system boundary of this LCA is set from gate to gate (Supplementary Fig. 24). The cut-off method was introduced into PET recycling to simplify the analysis. The functional unit is 1kg PET flake after sorting and shredding. The production process data of tap water, electricity, steam, and basic chemicals were obtained from ecoinvent v3.10 database. Fig. 4c shows that compared to direct incineration or landfill treatment of PET (2.61 kg CO₂-eq/kg PET)^{3,49,50}, converting waste PET into chemicals can greatly reduce greenhouse gas emissions and non-renewable energy use. Moreover, TPA and GA recovered from waste PET can avoid the significant consumption of fossil resources and a large amount greenhouse gas emissions caused by conventional production. As a result, the oxidation strategy for treating 1kg of waste PET can avoid the use of 37.09 MJ of fossil sources and reduce the emission of 2.25 kg CO₂-eq. In addition, the NREU and GWP recovered by one-step oxidation route are reduced by 15.09 MJ/kg PET and 0.91 kg CO₂-eq/kg PET compared to the alkaline hydrolysis process.” [Page 17, Line 329-341]

3. Results and discussion: The authors claim that the first dehydrogenation of EG to the corresponding aldehyde is the rate-limiting step. This should be verified by KIE, that can easily be performed.

Response:

Thank you for your comment. It is very effective and necessary to use KIE experiment to verify that C-H bond cleavage is the rate determining step in the reaction process.

Based on your suggestion, we performed the KIE experiment for EG and deuterated ethylene glycol (C₂D₄(OH)₂) oxidation. Supplementary Fig 20 shows that ethylene glycol (K_H=0.00805) reacts more rapidly compared to deuterated ethylene glycol (K_D=0.00229) by measuring the initial reaction rates, and the K_H/K_D=3.52 is calculated through parallel experiments. The above results demonstrate that the cleavage of C-H bond is the rate

determining step in the oxidation process of ethylene glycol. This conclusion is consistent with the DFT calculation results and has also been confirmed by other researchers [Science 2010, 330, 74-78; Sci. Bull. 2019, 64(23): 1764-1772; ACS Catal. 2020, 10, 3832–3837]. We have added relevant discussions in the revised manuscript.

Supplementary Fig 20. a) Reactant concentration as a function of reaction time and b) kinetic isotope effect in oxidation of $C_2H_4(OH)_2$ and $C_2D_4(OH)_2$.

Reaction conditions: 20mL 0.2M glycol , 50mg catalyst, 1Mpa O_2 , 80°C.

“Next, the C-H bond activation of the oxygen-containing intermediate is further promoted by the interfacial structure. Notably, this step is the rate-determining step (RDS) which was proven by the KIE (kinetic isotope effect) experiment (Supplementary Fig. 20) for EG oxidation, and Au/NiO-O_v exhibits a lower activation energy of RDS than Au/NiO.” [Page 14, Line 276-280]

4. Discussion: It would be worthwhile to compare the proposed method to other upcycling technologies for PET, such as:

Waste Manag. Res. 2009, 27 (8), 763–772

Resour. Conserv. Recycl. 2010, 54 (12), 1241–1249

J. Clean. Prod. 2019, 211, 1268–1283

ACS Sustainable Chem. Eng. 2024, 12, 4114–412

Response:

Thanks for your valuable suggestion. We further compare the Global Warming Potential

(GWP) and No-Rnewable Energy Use (NREU) values with the CO₂ emissions reported in the above literature to demonstrate the advantages of this work in energy conservation and emission reduction. We also further consulted more literature and analyzed the carbon emissions generated by various recycling processes. The negative sign denotes a decrease in carbon emissions and a reduction in the consumption of non-renewable resources.

Compared with landfill disposal (10.5kg CO₂-eq/kg PET), incinerating waste plastics for energy recovery can reduce the consumption of natural resources such as coal, thereby reducing the corresponding carbon dioxide emissions (1.4kg CO₂-eq/kg PET). However, the incineration process produces toxic gases like dioxane, which pollute the atmosphere and harm human health. As the main recycling method at present, mechanical treatment uses physical means to reuse plastics, which undoubtedly reduces the environmental pollution caused by waste plastics. By processing plastic waste received at the source through material recovery facilities (MRF), granulating it for downstream use, and extending the plastic life cycle to a certain extent. According to calculations by Thomas Astruop^[1] and Tom Chilton^[2], collecting and mechanically processing 1kg PET from the source can reduce 1.2~1.7kg CO₂ emissions, which is affected by the degree of industrialization in the local area. The emerging chemical recycling can achieve the circular and high-value utilization of PET. The production of BTX chemical raw materials by thermal cracking of PET can save 18.1MJ of non-renewable resources^[8]. The production of dimethyl terephthalate (DMT) from methanol-treated waste PET can reduce 1.88kg CO₂-eq^[9]. More and more chemical recycling^[9] routes are being developed and utilized. We adopt a one-pot method to depolymerize oxidized waste PET to prepare glycolic acid, which can reduce 37.09MJ of non-renewable resource consumption and 2.25kg of carbon dioxide emissions when treating 1kg of PET. Compared with the existing chemical recycling paths, this process has great advantages in energy conservation and emission reduction. We have added relevant discussions in the revised manuscript.

“Moreover, in comparison to enzymatic catalysis, the one-step method offers a reduction in emissions, attributed to its higher yield of glycolic acid and simpler separation process. Meanwhile, we also compared the depolymerization oxidation process with other PET recycling processes, including glycolysis pyrolysis and methanolysis (Supplementary Table 10), and the

results are gratifying. It is found that the depolymerization oxidation process for preparing glycolic acid is superior to the existing recycling processes in terms of energy saving and emission reduction.” [Page 17, Line 343-347]

Supplementary Table 10. Comparison of Carbon Dioxide Emissions from Different Recycling Routes.

Number	Recycling method	Product	NREU (MJ/kg plastic)	GWP (kg CO ₂ -eq/kg plastic)
1	Mechanical recycling	Recycled plastic	--	-1.2
2	Mechanical recycling	Plastic bottle	--	-1.7
	Burning	Electricity	--	1.4
3	Landfilling	Solid waste	5.65	10.5
	Incineration	Energy	-10.8	4.94
4	Methanolysis	DMT+EG	--	-1.88
5	Alkaline hydrolysis	TPA+EG	--	-1.17
6	Acetolysis	TPA+EDGA	70	2.19
7	Glycolysis	BHET	66	3.66
		TEA	3.2	0.40
		BTX	-18.1	1.88
9	This work	TPA+GA	-37.09	-2.25

[1] Waste Manag. Res. 2009, 27 (8), 763–772.

[2] Resour. Conserv. Recycl. 2010, 54 (12), 1241–1249.

[3] J. Clean. Prod. 2019, 211, 1268–1283.

[4] ACS Sustain Chem. Eng. 2024, 12, 4114–412.

[5] Chem. Eng. J. 2023, 470, 1385-8947.

[6] Nat Commun. 2023,14, 3249

[7] Resour, Conserv. Recycl. 2010, 55 (1), 34-52

[8] Energy Environ. Sci. 2023, 16, 3638–3653

Again, thanks very much for the reviewer’s careful review and valuable suggestions.

Reviewer #2:

The article by Chen and collaborators presents a significant advancement in the field, describing a one-step tandem strategy to achieve the thermal catalytic oxidation upcycling of PET to terephthalic acid (TPA) and high-value glycolic acid (GA) over a well-designed Au/NiO catalyst with rich oxygen vacancies. The method, which has been successfully applied to upcycle other polyethylene glycol esters to GA, demonstrates excellent catalytic performance. The manuscript and the results reported are not only interesting but also hold great potential for publication in Nat Comm after addressing the following points.

Major concerns:

1. On lines 188-190 and line 268 the author supposed “Inferentially, the O_v plays a crucial role in promoting EG oxidation against the inhibitory effect of TPA during the catalytic oxidation upcycling of PET (Fig. 2f)” and “TPA generated together with EG can preferentially adsorb on the oxygen vacancies of NiO support”. It is recommended to conduct the in-situ IR characterization with TPA addition to verify these conclusions.

Response:

Based on this suggestion, we conducted corresponding in-situ IR experiments to further verify the adsorption strength of EG and TPA produced by the alkaline depolymerization of PET on the catalyst surface.

Firstly, we performed in-situ IR to investigate the adsorption of EG and TPA on the catalyst. After the introduction of the corresponding solution, clear absorption peaks appear on the spectrum. This is due to the absorption of -OH group in ethylene glycol at 3300cm^{-1} and 1445cm^{-1} , while the absorption of -C=O and -COOH in terephthalic acid occurs at 1685cm^{-1} and 3260cm^{-1} wavelengths. As the reaction temperature increases from 50°C to 400°C , EG and TPA gradually detach from the catalyst under N_2 blowing. Notably, the EG and TPA substrate exhibits a higher dissociation temperature on Au/NiO- O_v than that on Au/NiO, indicating that the substrate has stronger adsorption capacity on the catalyst surface (Supplementary Fig. 16). Obviously, the vacancies on the NiO surface enhance the adsorption of ethylene glycol and terephthalic acid on the catalyst, which is consistent with theoretical calculations.

Supplementary Figure 16. In-situ IR spectra of different substrates adsorbed on catalyst surfaces.

Furthermore, FTIR was used to test the adsorption of substrates by Au/NiO and Au/NiO-Ov in a mixed solution (0.2M TPA, 0.2M EG, 0.5M NaOH). It is evident that the -OH absorption peak of EG located at 3450cm^{-1} increases significantly on Au/NiO-Ov compared to Au/NiO, while the intensity of -C=O and benzene ring absorption peak of terephthalic acid (TPA) located at 1559cm^{-1} ; 1457cm^{-1} and 1409cm^{-1} is relatively weak (Supplementary Fig. 17). This directly proves that the introduction of oxygen vacancies (Ov) increases the adsorption of EG on the catalyst surface, thereby facilitating the rapid conversion of ethylene glycol.

In summary, we have confirmed through in situ IR and FTIR that the presence of vacancies is more conducive to the adsorption of EG at the reactive sites, thereby accelerating the reaction rate and achieving efficient conversion of PET to glycolic acid.

Supplementary Figure 17. FTIR spectra of adsorption of EG and TPA on catalyst surfaces.

Detailed testing process of In-situ IR: The fourier transform infrared spectrum was measured on the Thermo Scientific Nicolet iS50 FTIR. The samples were pretreated by 50mL/min of N_2 at 200°C for 1h. After the sample cools to 30°C , scan and record the sample background. Then add 50ul of the corresponding solution to the in-situ cell. Set the heating program and scan the corresponding sample infrared spectrum under 25ml/min N_2 after reaching the corresponding temperature.

Detailed testing process of FTIR: Take 10ml of mixed solution (0.2M TPA, 0.2M EG, 0.5M NaOH) and 0.5g of catalyst (Au/NiO or Au/NiO- O_v), stir at room temperature for 4 hours, and then freeze dry the filtered catalyst. Press 0.1g KBr tablets and scan the spectrum as background. Subsequently, 0.01g of dried catalyst and 0.1g of KBr were mixed evenly and pressed into tablets. The spectra were scanned 64 times with a resolution of 4cm^{-1} .

“To further corroborate computational findings, a series of infrared (IR) tests were conducted to confirm that the introduction of O_v improves the adsorption of EG at the Au/NiO interface. Supplementary Fig 16 clearly shows that EG and TPA desorb at higher temperatures on the Au/NiO- O_v surface compared to the Au/NiO surface with vacancies, confirming that the presence of vacancies enhances substrate adsorption on the catalyst surface. By comparing the adsorption behaviors of the two catalysts in actual reaction solutions (Supplementary Fig. 17), it is apparent that Au/NiO- O_v adsorbs more ethylene glycol, which results in a faster conversion of EG to glycolic acid at the reaction's active sites. Consequently, we have substantiated our prior hypothesis through kinetic experiments, DFT calculations, and infrared analyses, that O_v diminishes the inhibitory impact of TPA on EG oxidation, mediated by the differential adsorption strength of the substrate on O_v .” [Page 11-12, Line 237-246]

2. On lines 263-265, the author supposed “Subsequently, the as-formed ethanol aldehyde is attacked by hydroxyl radical to generate $\text{CH}_2\text{OHCHOOH}$, which further undergoes dehydrogenation reaction to obtain GA.” It is recommended to conduct an in-situ EPR test on the role of hydroxyl radicals and check whether there are other organic radical intermediates involved in the formation of GA.

Response:

Thank you for your valuable suggestion. We also believe that it is necessary to use in-situ EPR to detect the effect of hydroxyl radicals in the reaction.

Firstly, we conducted a free radical quenching experiment by adding a hydroxyl radical quencher (salicylic acid) to the reaction, which results in a significant decrease in the yield of glycolic acid. On the contrary, the addition of a superoxide radical quencher (p-benzoquinone) resulted in almost unchanged reaction results, ruling out the influence of superoxide radicals on the reaction (supplementary table 4). Furthermore, we utilized in-situ EPR to detect the role of

hydroxyl radicals during the reaction process, and the results are shown in the following figure (Supplementary Fig. 21-22). No signal of hydroxyl radicals is detected before the reaction, but as the reaction progresses, a clear signal of hydroxyl radicals appears. The experimental results are consistent with the theoretical calculations, proving that the intermediate (ethanal) in the reaction is transformed into glycolic acid under the attack of free radicals. This result has also been confirmed by other researchers [Science 2010, 330, 74-78; Appl. Catal. B 2021 284 119803; Chemical Engineering Science 203 (2019) 228 236; ACS Catal. 2020, 10, 3832–3837]. We have added the relevant discussions into the revised manuscript.

Supplementary Table 4. Ethanol aldehyde oxidation under the condition of adding quencher.

Number	Reaction conditions	GA Yield
1	Glycolaldehyde Dimer	12.31%
2	Glycolaldehyde Dimer & Salicylic acid	5.18%
3	Glycolaldehyde Dimer & Benzoquinone	11.98%

Reaction condition: 20ml 0.15M glycolaldehyde dimer, 0.8g NaOH, 0.1g Au/NiO-400 100°C-10min.

Supplementary Figure 21. a) In-situ EPR spectra with free radical trapping agent (DMPO) for the oxidation of glycerol on Au/NiO-O_v (Reaction condition: 20ml 0.15M glycolaldehyde dimer, 0.8g NaOH, 0.1g Au/NiO-O_v, 100°C); b) Yield of ghanol acid at different reaction times.

Supplementary Figure 22. In-situ EPR testing process.

Detailed testing process of In-situ EPR: Firstly, add the reactants to a 50ml three necked flask and use a heating mantle for heating. During this process, pure oxygen is introduced into the reactor. When the corresponding reaction time is reached, quickly take out 1ml of the reaction solution and add 100ul of DMPO (5,5-Dimethyl-1-Pyrroline-N-oxide) to capture the hydroxyl radicals in the solution. Prepare liquid samples using capillary tubes to detect the intensity of hydroxyl radicals. The EPR spectra were collected between 3444 and 3544G in 100ms. The microwave frequency was 9.7GHz with a power of 0.2mW, and the field was modulated at 100kHz and with an amplitude of 5G.

“Subsequently, the as-formed ethanol aldehyde is attacked by hydroxyl radical to generate $CH_2OHCHOOH$, which further undergoes dehydrogenation reaction to obtain GA. This process was demonstrated through free radical quenching experiments and in-situ EPR characterization (Supplementary table 4 and Supplementary Fig 21 and 22).” [Page 14, Line 284-287]

3. The 1g-rated experiment may not be sufficient to demonstrate the potential of the upgrade from PET to TPA and GA. Thus, it is crucial to conduct a larger experiment or use flowing air or O₂ to illustrate the sustainability and scalability of this process.

Response:

According to your comment, it is difficult to verify the scalability of the process with a 1g scale experiment, so we conducted a scaled-up experiment.

Supplementary Figure. 4. The reaction and separation process of 30g PET.

In brief, 30g PET was added to a 500ml high-pressure reactor and reacted at 150°C for 3 hours. After removing the catalyst and unreacted PET, the liquid-phase yield of glycolic acid reached 85.62%, as determined by high-performance liquid chromatography. Following separation and purification, 8.95g of glycolic acid crystals were obtained, representing a yield of 75.71%. This demonstrates that the process of preparing glycolic acid by depolymerizing oxidized PET in one step possesses the potential for large-scale application.

In addition, when air is flushed in, the catalytic performance is much lower than that under pure oxygen. Although PET is completely depolymerized, the yield of glycolic acid is only 23.57%. The results indicate that a pure oxygen atmosphere is extremely favorable for the reaction.

Supplementary Table 11. Reaction results under different atmospheres.

Number	Atmosphere	TPA yield	EG yield	GA yield
1	1MPa O ₂	99.43%	3.21%	75.71%
2	1MPa Air	99.01%	66.93%	23.57%

Reaction condition: 300ml deionized water, 15g NaOH, 30g PET, 1g 1% Au/NiO-O_v 150°C-3min-500rpm.

“In addition, we conducted further scaling experiments to verify the feasibility of large-scale development. A 500ml high-pressure reactor was used to process 30g PET particles at once, and PET was completely reacted. After separation and extraction, 8.95g of ethanol acid (yield: 75.71%) and 25.78g of terephthalic acid (yield: 99.31%) were finally obtained (Supplementary Fig. 4).” [Page 5, Line 105-109]

4. Catalytic stability is very important for real applications. I am intrigued to see if the catalyst can consistently deliver its catalytic performance over 3-5 cycles.

Response:

Based on your suggestion, we conducted multiple catalyst cycling experiments and regenerated the catalyst. It is found that Au/NiO exhibits excellent recycling and regeneration abilities.

Supplementary Figure 14 shows that the reaction performance of Au/NiO-O_v decreases with increasing reaction cycles. It is highly possible that the oxygen vacancies on the catalyst are gradually consumed as the number of cycles increases. Notably, by calcining the catalyst under an inert atmosphere (N₂), it can be observed that the reaction performance is restored, which may be due to the regeneration of oxygen vacancies during high-temperature treatment. To verify this conjecture, we utilized electron paramagnetic resonance (EPR) to measure the oxygen vacancies of the catalyst before and after the reaction. As shown in Supplementary Fig 14, the Au/NiO catalyst contained more oxygen vacancies before the reaction (6.831×10^{-5} mmol/g). However, after four cycles, the oxygen vacancy signal of the catalyst on the EPR spectrum significantly decreased. Subsequently, calcination of the catalyst in a nitrogen stream generated more vacancies, the oxygen vacancy signal was enhanced, and the concentration was restored to the pre-reaction level (7.095×10^{-5} mmol/g). Obviously, calcination under an inert atmosphere restores the activity of the catalys. This further proves that oxygen vacancies play a crucial role in the catalytic reaction process.

Supplementary Figure 14. a) The catalytic effect of Au/NiO-400 in mixed solutions with different ratios of ethylene glycol and terephthalic acid. (20ml 0.1M ethylene glycol solution, 0.1g catalyst, 0.4g NaOH, 1MPa O₂, 80°C, 10min); b): Catalytic stability of the Au/NiO-400 under multiple cycle test conditions (1g PET, 20 ml H₂O, 0.8g NaOH, 0.1g catalyst, 130°C, 3h); c) EPR spectra of catalysts under different cycle numbers; d) Oxygen vacancy concentration -- EPR simulation value.

“Moreover, the catalytic activity of Au/NiO-400 gradually decreased as the number of catalytic cycles increased (Supplementary Fig. 14). The yield of glycolic acid decreased by 37.2% compared to that of the fresh catalyst after four cycles. After calcining the spent catalyst in a nitrogen atmosphere, the catalytic activity of Au/NiO-400 was recovered due to an increase in surface oxygen vacancies. However, due to aggregation of gold particles, the glycolic acid yield was lower than that reacted with the fresher catalyst.” [Page 11, Line 213-218]

5. The ref part contains some minor mistakes. For example, “Acs Catal” should be written as “ACS Catal”. Yan et.al

Response:

Thank you for pointing out our mistakes in this regard, which may have caused ambiguity. We have promptly corrected the formatting issues in the references.

Revise as follows:

24. Kim, H.T. et al. Biological Valorization of Poly(ethylene terephthalate) Monomers for Upcycling Waste PET. *ACS Sustainable Chemistry & Engineering* **7**, 19396-19406 (2019).
-
35. Liu, N. et al. Au^{δ-}-Ov-Ti³⁺ Interfacial Site: Catalytic Active Center toward Low Temperature Water Gas Shift Reaction. *ACS Catalysis* **9**, 2707-2717 (2019).
36. Estellé, J. et al. Comparative study of the morphology and surface properties of nickel oxide prepared from different precursors. *Solid State Ionics* **156**, 233-243 (2003).
37. Steimecke, M. et al. Higher-Valent Nickel Oxides with Improved Oxygen Evolution Activity and Stability in Alkaline Media Prepared by High-Temperature Treatment of Ni(OH)₂. *ACS Catalysis* **10**, 3595-3603 (2020).
38. Mochizuki, C. et al. Defective NiO as a Stabilizer for Au Single-Atom Catalysts. *ACS Catalysis* **12**, 6149-6158 (2022).
-
46. Wang, H. et al. Electrocatalysis of Ethylene Glycol Oxidation on Bare and BiModified Pd Concave Nanocubes in Alkaline Solution: An Interfacial Infrared Spectroscopic Investigation. *ACS Catalysis* **7**, 2033-2041 (2017).
-
49. Zhou, X. et al. Glycolic Acid Production from Ethylene Glycol via Sustainable Biomass Energy: Integrated Conceptual Process Design and Comparative Techno-economic-Society-Environment Analysis. *ACS Sustainable Chemistry & Engineering* **9**, 10948-10962 (2021).
50. Muangmeesri, S. et al. Recycling of Polyesters by Organocatalyzed Methanolysis Depolymerization: Environmental Sustainability Evaluated by Life Cycle Assessment Analysis. *Acs Sustainable Chemistry & Engineering* **9**, 4114-4120 (2024).

Thank you to all the reviewers for your recognition of this work. Based on your valuable suggestions, we have added relevant experiments to make the manuscript more in line with the requirements of nature communication. This helps us gain a deeper and more comprehensive understanding of chemical recycling of waste polymers, and we look forward to receiving your reply.

Responses to Reviewers' Comments

Dear Reviewers,

We are very happy that the article was accepted by nature communications. Thank you for your valuable suggestions on the article during the publication of this article, so that the article can be improved and finally accepted. At the same time, we have completed the corresponding revision to the suggestions put forward again by experts. We hope that these amendments are satisfactory.

Reviewer #1:

The authors have taken great care in revising the manuscript and this reviewer supports publication.

Again, thanks very much for the reviewer's careful review and valuable suggestions.

Reviewer #2:

In this revised version, the authors have addressed most of my concerns and points related to the catalytic role and the structure-performance relationship of the Au/NiO-Ov. I only have two minor suggestions for this version, and have no doubt in recommending the publication after minor revision.

Major concerns:

1. There is no spectra of pristine catalyst when the authors conducted FTIR spectra of adsorption of EG and TPA on catalyst surfaces. It should be added for better comparison.

Response:

Based on this suggestion, We supplemented the FTIR data of the pristine catalyst to better compare the adsorption differences of TPA and EG on the catalyst surface

Supplementary Figure 17. FTIR spectra of adsorption of EG and TPA on catalyst surfaces.

Detailed testing process of FTIR: Take 10ml of mixed solution (0.2M TPA, 0.2M EG, 0.5M NaOH) and 0.5g of catalyst (Au/NiO or Au/NiO-O_v), stir at room temperature for 4 hours, and then freeze dry the filtered catalyst. Press 0.1g KBr tablets and scan the spectrum as background.

Subsequently, 0.01g of dried catalyst and 0.1g of KBr were mixed evenly and pressed into tablets. The spectra were scanned 64 times with a resolution of 4cm^{-1} . Au/NiO and Au/NiO-Ov were also scanned to provide significant adsorption contrast.

2. It should not be named as "in-situ EPR" for their EPR characterization. From their description, it is better to be called "ex situ".

Response:

Thank you for your valuable suggestion. The expression of "in-situ EPR" is indeed inaccurate, and we have all modified it to "ex-situ EPR".

Supplementary Figure 21. a) Ex-situ EPR spectra with free radical trapping agent (DMPO) for the oxidation of glycerol on Au/NiO-O_v (Reaction condition: 20ml 0.15M glycolaldehyde dimer, 0.8g NaOH, 0.1g Au/NiO-O_v 100°C); b) Yield of ghanol acid at different reaction times.

Supplementary Figure 22. In-situ EPR testing process.

Detailed testing process of Ex-situ EPR: Firstly, add the reactants to a 50ml three necked flask and use a heating mantle for heating. During this process, pure oxygen is introduced into the reactor. When the corresponding reaction time is reached, quickly take out 1ml of the reaction solution and add 100ul of DMPO (5,5-Dimethyl-1-Pyrroline-N-oxide) to capture the hydroxyl radicals in the solution. Prepare liquid samples using capillary tubes to detect the intensity of hydroxyl radicals. The EPR spectra were collected between 3444 and 3544G in 100ms. The microwave frequency was 9.7GHz with a power of 0.2mW, and the field was modulated at 100kHz and with an amplitude of 5G.

Thank you to all the reviewers for your recognition of this work. Based on your valuable suggestions, we have added relevant experiments to make the manuscript more in line with the requirements of nature communication.